# The eSports Medicine: Pre-Participation Screening and Injuries Management—An Update

**DOI:** 10.3390/sports11020034

**Published:** 2023-02-01

**Authors:** Agostino Rossoni, Marco Vecchiato, Erica Brugin, Eliana Tranchita, Paolo Emilio Adami, Manuela Bartesaghi, Elena Cavarretta, Stefano Palermi

**Affiliations:** 1Laboratory of Clinical Physiology and Sports Medicine, Department of Experimental Medicine, University of Milano-Bicocca, 20900 Monza, Italy; 2Sports and Exercise Medicine Division, Department of Medicine, University Hospital of Padova, 3651 Padova, Italy; 3Cardiovascular Rehabilitation and Sports Medicine Service, 30033 Noale, Italy; 4Department of Movement, Human and Health Sciences, University of Rome “Foro Italico”, 00135 Rome, Italy; 5Department of Cardiac Surgery, Cardiology and Heart Lung Transplant, Bambino Gesù Children’s Hospital, IRCCS, 00165 Rome, Italy; 6Health and Science Department, World Athletics, 98007 Monaco, Monaco; 7Laboratoire Motricité Humaine Expertise Sport Santé (LAMHESS), Université Côte d’Azur, 06000 Nice, France; 8Department of Medical-Surgical Sciences and Biotechnologies, Sapienza University of Rome, 04100 Latina, Italy; 9Mediterranea Cardiocentro, 80122 Napoli, Italy; 10Public Health Department, University of Naples Federico II, Via Pansini 5, 80131 Naples, Italy

**Keywords:** electronic sport, eSports, eGames, eAthletes, sports medicine, cardiovascular prevention, injury prevention

## Abstract

Recently, electronic sports (eSports) became one of the growing forms of new media due to the wide diffusion of games and online technologies. Even if there is still a debate about the definition and characterization of eSports, eAthletes train heavily, compete in tournaments, must abide by competition, association, and governing body rules, just like all other athletes. Furthermore, as in any other competitive discipline, there can be injuries. Aberrant sitting posture, repetitive movements, screen vision, prolonged playing hours, and a sedentary lifestyle can lead to several medical hazards in musculoskeletal, ophthalmology, neurological, and metabolic systems. Moreover, several cardiovascular changes occur in eAthletes. This paper aims to explore the different injuries that can occur in a professional eAthlete, suggesting how every high-level gamer could benefit from a pre-participation evaluation and a correct injury prevention strategy.

## 1. Introduction

Even if a recognized definition is still missing, electronic sport (eSport) has been traditionally described as “a form of sport where the primary aspects of the sport are facilitated by electronic systems, and the input of players and teams are mediated by human-computer interfaces” [1].

During recent years, especially during the COVID-19 pandemic period, eSports became rapidly a growing form of new media thanks to the increasing availability of online games and broadcasting technologies. Some events and tournaments culminate in championships at the regional and international levels, in which players compete against each other, as single players or teams [2,3,4]. The phenomenon of eSport attracts many spectators: nowadays, people regularly consume eSports events through online-developed streaming platforms [5] (YouTube, Twitch, or Smashcast), gaining a global audience of about half of a billion fans and passing the one billion dollars mark [6].

ESport, as a sport discipline, is growing in recognition worldwide. In China, it has been already defined as a sports activity since the beginning of the 2000s and will be an official competitive sports event in the 19th Asian Games in Hangzhou 2023 [7]; in the USA, eAthletes have been considered professional players since 2013. The phenomenon is so widespread that the International Olympic Committee (IOC) accepted eSports as a sport in 2017 and there have been speculations about their introduction in the Paris 2024 Olympic Games [3,8]. Moreover, in 2021, IOC created the Olympic Virtual Series, composed by baseball, cycling, rowing and motor racing [9].

Even if e-sport has received a wide international acknowledgment, there is still resistance as to whether it can truly be considered as a sport discipline played by real athletes, although it is undoubtful that this discipline has a big impact on players’ health. Indeed, several reports have been published [10,11] and this is a growing topic in the scientific literature.

Therefore, this paper aims to review the numerous different injuries that may occur in these athletes, named eAthletes, trying to highlight the potential benefit of a pre-participation evaluation and a correct injury prevention program.

## 2. Can We Consider ESport as a Physical Sport Discipline?

Before eSport could be classified as a professional sport, two main criteria related to the concept of sports should be explored [12]. The first one is about physical performance and the skillful use of the body or a part of it, similar to what happens in some sports (i.e., in shooting only arms are used). For the second criterion, eSports require institutional stability, with rules for regulation and governance. Even if the different types of eSports make it difficult to achieve universal institutional stability, global eSports organizations already exist, such as the International eSports Federation [13]

There still is no general agreement about the classification of eSports, but they can be differentiated into specific genres such as multiplayer online battlefield arena (MOBA), first-person shooters (FPS), real-time strategy (RTS), or sports simulations [5]. Others classified them as shooter, band simulation, dance simulation, and fitness categories [7]. In every genre, different digital games can be found, and played through various platforms (e.g., computers, consoles, mobile, or virtual reality), with different mechanics and competition rules [14]. All these different modalities of play lead obviously to different types and severity of health problems.

As for other types of sports, we can find professional eSports players and amateur players. Professional players in eSports share many similarities with their traditional physical athlete counterparts: they train rigorously, compete in tournaments, and must abide by competition, association, and governing body rules [15]. They have the same mental and physical issues as physical athletes, sharing some concerns such as stress before competitions or short career life. The eAthlete signs a contract with a professional e-team, receiving salaries and sponsorship; moreover, the infrastructure of eSports can be compared to those of traditional physical team sports [6].

Nowadays professional clubs from different sports such as football, basketball, and F1 racing are increasingly getting involved in eSports to extend their recognition, setting up their e-teams to compete for national and international titles. Scientific literature described several benefits of eSports [7,16,17]:No gender differencesImprovement of the health and well-being of sedentary individualsGroup participation and competition regardless of locationPromotion of physical activity for individuals with physical and cognitive disabilitiesEnhancement of cognitive development and school achievementEnhancement of social interactionImprovement of fine motor skills and visual processingPromotion of an effective rehabilitation.

However, some harmful effects are still described [7,16,17]:Potential treatments to child safetyInappropriate contentExposure to violenceBullyingInternet addictionReduction of moderate/vigorous physical activityDisplacement from active social life and interactionsExposure to junk food advertisingSleep deprivationVision complaintsMusculoskeletal overuse.

To achieve top performance in an eSports game, professional eAthletes train on average 5 h per day to improve specific game skills and motor abilities, such as gamepad and keyboard handling, eye-hand coordination, reaction time [18], game knowledge as well as strategy and tactics; during competition, they must be able to sustain high levels of attention and take important decisions under time pressure [6].

## 3. ESports Health Problems

As for training, medical problems of eAthletes are still scarcely reported in the scientific literature [19] and a systematic analysis of it is still missing in current literature. Nevertheless, scientific research suggests that gaming may have several undesirable physical and mental health effects (Table 1) [11,20]; in the study by Di Francisco-Donoghue et al. [4] eAthletes reported that about half of the players suffered from eye fatigue, neck and back pain, and wrist and hand pain. Therefore, prevention, diagnosis, and treatment of eSports-related issues are needed. 

### 3.1. Musculoskeletal System

To achieve top performance, eAthletes spend several hours per day with repetitive movements using a gamepad, mouse, and keyboard and sitting for an extended period often with sub-optimal posture: these are the two major mechanisms causing musculoskeletal complaints in this sport [21]. Indeed, different platforms of play have important consequences on gamers’ posture [22,23]. Neck and back pain are the two most common problems, even if other body districts may be interested [4]. 

The typical forward displacement of the head while playing leads to anomalies in the cervical spine and muscles, possibly causing repercussions also to the lower back: early degeneration in neural disks and roots, peripheral compressions, and radiculopathies are frequent in eAthletes [21,24]. Appropriate interventions for eAthletes with back pain include exercise therapy, stretching and manual manipulation, together with physical therapies [25], to reduce muscle tension and improve mobility and flexion of the spine [26]. Stretching and strengthening should focus on muscles associated with postural control [27]. Mezieres and McKenzie are two used methods for it.

Competitive eAthletes engage in fine motor movements, with both isotonic and isometric fine contractions: they execute more than 300 high-task movements per minute, predisposing to repetitive and chronic injuries [28,29]. Hand and wrist pain issues are reported by 30% of players [28,29]. Carpal or ulnar tunnel syndromes, De Quervain’s tendinopathy, and elbow pain are common diagnoses.

Therefore, a musculoskeletal screening examination is mandatory for eAthletes, also using methods already described [30]. The use of diagnostic instruments is helpful, such as the X scan for cervical spine alignment, magnetic resonance imaging for lumbar herniation, electromyography for radiculopathy [31], and ultrasound for muscle, ligament, and tendon injuries [25].

There are several ergonomics assessment tools available, developed by organizations such as the U.S. National Institute of Occupational Safety and Health, that can be used in screening musculoskeletal and ergonomic complaints in these athletes. Monitor, chair, desktop, keyboard-mouse-controller, and lightning should be the main ergonomics to assess.

### 3.2. Ophthalmology System

ESports also require visual and attentional stamina [32]. Recently, vision issues likened to the use of digital screens have become a huge public health problem. “Digital eye strain” is a newborn term created to include all aspects of eyesight issues related to long working sessions in front of a digital screen [33]. EAthletes require to have their eyes fixed on a computer screen with excessive exposure to light-emitting diodes [8], and this can impact the natural circadian rhythm [34], especially during tournaments, which might last hours. Moreover, screen time before sleeping has been linked with “screen insomnia,” a common among gamers [35]. Recently it has been demonstrated that excessive exposure to blue light spectrum can create retinal and photoreceptor problems [36]. 

Limiting playtime or using particular devices (e.g., screen glasses) to avoid these hazards should be the main aim in the medical management of these athletes. The American Optometric Association suggests the 20/20/20 rule, taking 20 s of break every 20 min to view something 20 feet away (i.e., take a 20-s break to view something 20 feet away every 20 min) [37].

Assess the visual acuity, evaluate position and alignment of the eyes, and assess extraocular movements, perform the Vestibular Ocular Motor Screen, test saccades and assess convergence and accommodation amplitude should be incorporated into the evaluation of these athletes. Oculomotor performance training can optimize performance [7]: eye exercises are a key component of this strategy.

### 3.3. Nutrition

Athletes, to compete at their optimum level, need to pay attention to their nutrition; according to the sports discipline, they may require specific calories, macronutrients, and micronutrients to maintain strength and energy and ensure adequate hydration. However, it has been recently demonstrated how eAthletes often have bad nutritional habits [38]: their meals are irregularly consumed, snacking is frequent, and sugared and fried foods are commonly eaten. These unbalanced and largely unhealthy dietary habits raise several health concerns.

A balanced diet is crucial for eAthletes, mainly belonging to a young age group (between 18–25 years) [4], who should need an intake from 1800 to 3000 Kcal/day, according to the World Health Organization [39]. Avoiding a sedentary lifestyle and following a personalized diet should be two main aims for the nutritional management of eAthletes [35]. Body mass index should be routinely evaluated for these athletes, other than nutrition habits using validated questionnaires [40,41]. The evaluation of fat mass and fat-free mass through bioelectrical impendence analysis or plicometry could also be useful. 

Inadequate intake of nutrients leads to deficiencies, possible causes of many diseases [42]: it is the cause of deficits of vitamin B12, folate, and zinc that may cause symptoms of fatigue, depression, and dementia. Furthermore, there is growing evidence that specific dietary supplements may enhance cognitive performance, such as caffeine, creatine, or polyphenols [43].

Energy drinks (EDs) are more commonly used by the young population as a source of dietary supplements, especially among eAthletes who spend many hours sitting, sometimes without a proper diet. The common ingredients of EDs are caffeine, sugar, taurine, and ginseng [44]. Caffeine has been classified as a supplement with benefits also in terms of sports performance. Lower doses may improve concentration and reaction times, while higher doses can influence cognitive and physical performance, also in eAthletes [43]. However, it has been demonstrated that caffeine can have pro-arrhythmic and blood pressure-increasing effects, especially when the dosage exceeds 300 mg per day or in susceptible individuals at much lower dosage [45]; other components of EDs can also contribute to cardiovascular side effects, ranging from benign arrhythmias to potentially fatal cardiovascular diseases. Furthermore, the sugar component of EDs induces insulin resistance, and stimulates proinflammatory activity causing dyslipidemia and weight gain [44]. Moreover, although caffeine is currently not on the World Anti-Doping Agency (WADA) List of Prohibited Substances, it is constantly monitored for its possible re-introduction.

Consequently, closer attention should be paid to eAthletes’ nutrition and supplementation, such as the use/abuse of some type of EDs, raising awareness about the positive and negative effects of these products. The ASSIST is a helpful screening questionnaire for monitoring players’ supplement use [46]. For eSports to be fully acknowledged and accepted among other sports disciplines, it will be necessary that doping control rules and regulations apply similarly as they do in other sports.

### 3.4. Neurological System and Mental Health

The effects of gaming on the human brain have been recently questioned in literature [47]. Insomnia or sleep deprivation, headache, and stress [48] are frequently found in eAthletes. Long hours of online gaming could be linked to the development of some mental issues such as depression, social phobia, obsession-compulsion disorder, and gaming addiction [48]. Indeed, despite differences in socio-demographic and playing variables exist, eSports and physical sports players tend to exhibit lots of similarities in psychological domains: therefore eSports psychology must not be underestimated, given the recognition of gaming disorder in ICD11 [49], and adequate mental health strategies are needed [50]. The use of some validated questionnaires can be useful to diagnose these problems [7]: the Patient Health Questionnaire (PHQ), followed by the PHQ-9 can screen for major depressive disorders; Generalized Anxiety Disorder (GAD) can be screened using the GAD-2 and GAD-7 questionnaires; Internet Gaming Disorder can be screened using the Internet Gaming Disorder Scale 9-Short Form. Cognitive training, relaxation exercise, and breathing techniques are crucial component of a correct mental injury management, in order to build strong mental skills [7]. 

Screens of digital devices may interfere with sleep: indeed, blue light suppresses melatonin production, affecting circadian rhythms and impairing cognitive performance [7]. This is a big problem since many eAthletes play during nighttime due to the longer uninterrupted duration [51]. The use of blue light-blocking glasses and a sleep hygiene plan can improve sleep duration and quality [36]. The Pittsburg Sleep Quality Index is a validated questionnaire that could be useful in sleep monitoring [52].

When using virtual reality devices, players may develop symptoms similar to motion sickness, the so-called cybersickness: the human brain must integrate real-time vision, hearing, vestibular and proprioceptive inputs, therefore resulting in a high-task role [53]. On the other side, however, there is evidence indicating how active video games have a positive psychological impact, concerning liking and enjoyment, attitudes, and self-efficacy [54]. Online social interaction, via eGames, has been found to strengthen meaningful feelings and self-regulation, improving some social needs otherwise. Moreover, gaming was shown to be effective also as a therapy against mental disorders, such as attention deficit hyperactivity disorder (ADHD) [55,56,57,58]. Recent interesting papers showed how playing Minecraft [59] or SuperMario [60] were associated with an improvement of hippocampal-associated memory [59].

Some recent and growing papers examined the influence of eSports experience on human development, showing a promising positive relationship between eSports and neural plasticity [61,62]. Moreover, the high degree of participation, the controlled environment, the organized skill ratings, the social nature, and the large amount of data, make the eSport a valid field for neurocognitive research [47]. The metaverse concept can be a promising field in that sense [63].

## 4. Physical Fitness

Despite the common point of view, eSport is not a passive activity. However, an eAthlete could spend a considerable amount of time in sedentary activities, and video games are a substitute for more intensive exercise because the level of effort achieved is not sufficient to improve cardiorespiratory fitness [64,65]. 

Indeed, a recent survey observed that almost 90% of elite eAthletes exercise regularly in the gym for about 5 h daily, even more than the physical activity recommendations of WHO for both children and adults [66,67]. Unfortunately, approximately half of the eAthletes investigated did not believe that introducing physical exercise in their training program could have a positive effect on their gaming performance, not recognizing the important relationship between physical and mental health [68]. Conversely, in a more recent survey, Di Francisco et al. reported that 40% of collegiate eAthletes do not participate in any form of physical activity, and Trotter et al. stated that 80% of eSports players do not meet the WHO’s physical activity recommendations [4,69]. 

Therefore, this is a controversial point in the scientific literature, and more studies are needed. “The e-sport study 2019”, a study about the demographics and health behavior of eSports players, showed that the professional eAthlete has excellent health and physical status: indeed, eAthletes ranked in the top 10% were more physically active compared to the other players [69].

It is important to screen the physical fitness of these eAthletes. Physical fitness comprises several aspects, including cardiorespiratory fitness (i.e., through cardiopulmonary exercise test with VO2 max evaluation [65,70]), muscle endurance and strength, body composition, balance, and flexibility: testing these is necessary to set a correct training plan, monitor progresses and evaluate physical functioning [71]. 

The eSports world has enormous potential for physical activity and health-promoting efforts, due to its broad dissemination to a wider public [72]. Indeed, several platforms have developed eGames where level advancement requires the player to walk, run, and ride a bicycle, mixing reality with the virtual world [73]. An interesting recent research shows how eSports could be considered as an exercise prescription in patients with cystic fibrosis [74], to promote physical activity in people with diseases: this is a very important finding as the use of physical activity as a drug against diseases in gaining growing scientific interest [68,75], and therefore eSports can play a role in that field, also though the digital diffusion of physical exercises [76]. Indeed, Tabacof et al. [77] showed how technology and adaptive competitive eSports have the potential to improve social connectedness and inclusion in people with quadriplegia. Moreover, these technologies have been used for other practical tasks such as the mental preparation of athletes for competitions, the creation of rehabilitation exercises for injured athletes or the achievement of some competency goals. Considering the high prevalence of physical inactivity worldwide and the latest diffusion of unhealthy habits also related to the COVID-19 pandemic [41], augmented reality mobile games may provide an enjoyable alternative to engage people who would otherwise never exercise, offering an attractive alternative to traditional physical exercise [78]. 

## 5. Cardiovascular System

Despite their growing scientific interest, the physiological demand for eSports has been poorly explored. Cardiovascular adaptations of eSports are surely one of the most debated topics in this context (Figure 1).

An increase in heart rate (HR), blood pressure (BP), and cardiorespiratory fitness during eSports gaming sessions have been observed [79,80]. Like any other skill discipline [81], eSports can cause the release of catecholamines and a consequent rise in BP by augmenting cardiac contractility and increasing HR [60]. The HR increase indicates a parasympathetic withdrawal and/or an upregulation of the sympathetic nervous system possibly through the release of catecholamines. This modulation in eAthletes is particularly evident in some game situations [82]: indeed, HR increases at the beginning of the match or toward the end of a game, and it is higher when the presence of a human opponent is compared with a computer opponent [83]. This means that the sympathetic nervous system in eSports players is activated by competitive play and is modulated by the game situation.

During Fortnite sessions, an eGame in which the player continues playing for as long as possible in an uninterrupted session with increasingly difficult waves of challenges, Valladão et al. [84] described a rise in the mean and peak HR, suggesting that an eAthlete experiences a stressful physiological response. 

Moreover, given the rise of systolic BP and HR while playing eSports, other cardiovascular alterations, such as arrhythmias, might be expected to occur in eAthletes. Claire M. Lawley et al. [85] described, in a correspondence published in the New England Journal of Medicine, four instances of syncope in three children due to ventricular tachycardia or ventricular fibrillation while playing electronic games. Mental stress and heightened emotion, such as anger, have been shown to shorten the cardiac ventricular action potential, thus triggering cardiac arrhythmias, in predisposed subjects, such as subjects suffering from long-QT syndrome [86].

Yamagata et al. [87] in their recent article hypothesized three different mechanisms causing the cardiovascular sequelae in eAthletes: sleep deprivation (linked to increased arterial stiffness and endothelial dysfunction, HR variability, cardiac chamber alteration, and cardiac repolarization and increased inflammatory markers), psychological stress (linked to cardiac arrhythmias, increased sympathetic tone) and stimulant use (linked to hypertension, increased level of circulating catecholamines, QT prolongation and coronary artery intimal hyperplasia).

Another potentially worrying cardiovascular aspect is caused by prolonged sitting time. Indeed, water retention can be a consequence of that, measured by bioelectrical impedance and reduced by wearing compression stockings [88]. Deep venous thrombosis is a rare but devastating consequence of that, as gaming has been linked to an hypercoagulability state [89,90]: a prompt diagnosis is crucial when a clinical suspicious raises, with compression ultrasound as a key diagnostic exam [91].

Cardiovascular safety during sports participation aims at avoiding hidden and potentially fatal cardiac events [92,93], through the use of pre-participation cardiovascular strategies, promoted by medical societies [92]. Sudden cardiac death in young athletes is caused by a variety of structural and electrical disorders of the heart, including cardiomyopathies, ion channel disorders, coronary anomalies, and acquired cardiac conditions [92]. Likely, some of the eSports players of all ages have either known or hidden cardiovascular disease. Even minor cardiovascular changes in these people might occasionally have serious consequences [94]. Because it is not feasible to evaluate the entire population that plays video games extensively or intensively, it may be useful to start with professional eAthletes. Therefore, the implementation of a pre-participation evaluation for eAthletes, similar to what is already practiced for skill disciplines athletes, should be considered. A medical evaluation investigating family and personal history, performing a comprehensive physical examination, and using a resting ECG, together with a resting BP value, may be appropriate to identify most cardiovascular conditions that may increase the risk of sudden death in young players. Echocardiography, Holter ECG and magnetic resonance imaging can play a role as a second and third line of athlete’s heart diagnosis [95].

## 6. Conclusions

The media attention and commercial interest in eSports are constantly increasing as well as the number of athletes participating in this discipline. The practice and dedication required for professional eSports are not different from that of any other skill-based discipline. At the same time, an effort must be made to emphasize the importance of practicing more active and intensive physical training accompanied by proper nutrition strategy to maintain eAthletes health. Therefore, there are potential clinical risks for the health of eAthletes associated with eSports practice that has recently become of medical interest. Every gamer who competes or aspires to compete at a high level should be evaluated through a complete pre-participation evaluation (Figure 2) and work on a correct training plan and an injury prevention strategy (Table 2) [96].

## 7. Future Directions

With the help of virtual reality, augmented reality, and artificial intelligence, among other technologies, traditional sports can follow successful strategies from interactive media [79]. Furthermore, specialized monitoring technologies and clinical grade assessments are becoming a topic of growing interest [96]. Future research areas of eSport medicine could be focused on musculoskeletal injuries, ergonomic optimization, vision screening, nutrition strength and conditioning, athlete longevity, performance assessment, and enhancement.

## Figures and Tables

**Figure 1 sports-11-00034-f001:**
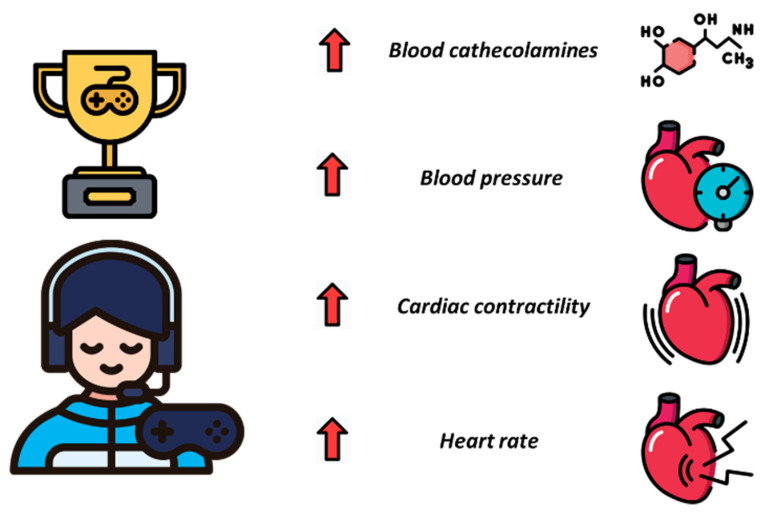
Acute cardiovascular response during eSports session.

**Figure 2 sports-11-00034-f002:**
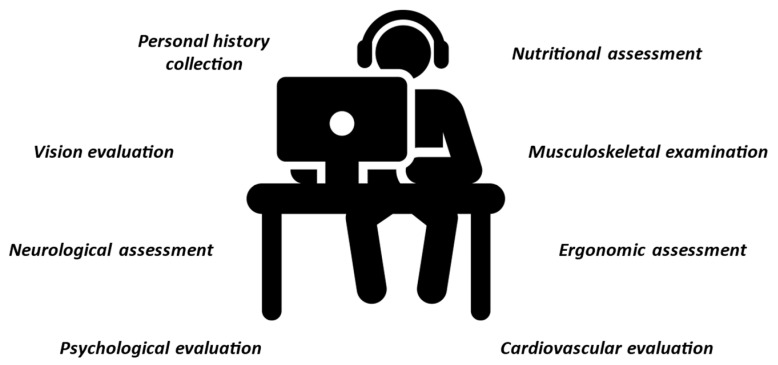
Pre-participation evaluation proposed model for eAthletes.

**Table 1 sports-11-00034-t001:** Health problems of eSports.

Body Part	Mechanism	Injury
Musculoskeletal system	Aberrant sitting prolonged posture	Neck pain
		“Text neck” syndrome
		Low-back pain
		Radiculopathy
		Gluteal pain
		Hamstring tightness
	Repetitive movements	Shoulder overuse tendinopathy
		Elbow overuse tendinopathy
		Wrist overuse tendinopathy
		Cubital tunnel syndrome
		Carpal tunnel syndrome
		“Gamer’s thumb”
		Microtrauma
Ocular system	Long-lasting screen vision	Eye fatigue
		Dry eyes
		Temporary alterations of visual functions in children
	Virtual reality headset gaming	Alteration in the ocular surface and tear film
	Vergence-accommodation conflict	Decreasing vergence ability and accommodative facility
	Increased neural plasticity	Increased visual attentional resources in multi-object tracking
Nutrition	Overeating	Weight increase, obesity
	Energy expenditure, metabolism	Sedentary lifestyle, higher body fat percentage
	Wrong alimentation	Nutrients’ deficiency
	Increasing use of caffeine and energy drinks	Cardiac arrhythmias
Neurological system	Long-lasting screen vision	Headache
		Stress
		Insomnia
	Virtual reality headset gaming	“Cybersickness”
Mental Health	Addiction, isolation, several playing hours per game	Gambling
		Depression
		Stress
		Other psychosocial problems

**Table 2 sports-11-00034-t002:** Injury prevention strategies for eAthletes.

Stretching routine before and during gaming
Strengthening exercise
Appropriate posture
Appropriate ergonomics
Limit continuous gaming
Specific eyewear
Maintain volumes in an acceptable range
Adequate level of physical fitness
Adequate level of caloric and water intake
No excessive use of caffeine and energy drinks
Prefer healthy social interactions and team building
Relaxation techniques and meditation
Anxiety and stress control

## Data Availability

Not applicable.

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
