# Peer review of "The eSports Medicine: Pre-Participation Screening and Injuries Management—An Update"

_sports, 2023, doi:10.3390/sports11020034_

Round 1

Reviewer 1 Report

This is an interesting narrative review mapping the health problems of e-sports athletes. The article framed different areas of health problems that summarize evidence and theory. There are some more important information that shall be supplement in the paper.

-          The formal term of esports shall be without hyphen (i.e., not e-sports).

-          In some of the terminology when the authors refer to “real sports”, please consider revise some of them as “physical sports”,  at place where applicable to convey a better meaning.

-          There is an existing paper on the definition and re-definition of esports, which worth discussing.

-          In the background, the authors need to review and acknowledge other esports review papers and their other area of interest.

-          In the last paragraph of Section 1, the authors are encouraged to brief the overall framework of the upcoming paragraphs to ease the readers.

-          Table 1 and Table 3 did not seem to be good “tables”. I recommend making it as a bullet-point list.

-          For Section 2, There is another commentary article that discussing the topic: “recognizing esports as sport”. Please consider extract and reference to some of the contents.

-          For Section 3, I recommend to change it as “health problems” instead of “clinical hazards”, in which some of the issues might not be that “clinical”.

-          The is one critical point missed in the review that must be highlighted in the review. That is the medium/platform of the esports. It could be computer/console, mobile phones, and X-reality (particular virtual reality). They manifest different playing postures and shall be clearly mapped in the health problems, especially the musculoskeletal system and eyes. For the concluding remarks, I recommend to discuss the implications of Virtual reality exergames, which in a sense that it is approaching or fusing to “physical sports”. Please consider reinforced the context by the following reference.

Miah, A., Fenton, A. and Chadwick, S., 2020. Virtual reality and sports: The rise of mixed, augmented, immersive, and esports experiences. In 21st Century Sports (pp. 249-262). Springer, Cham.

-          Line 109, the authors said that descriptive statistics on health problems of e-athletes are scarce. That is one worth mentioning and it dedicates to mobile phone esports.

Lam, W.K., Liu, R.T., Chen, B., Huang, X.Z., Yi, J. and Wong, D.W.C., 2022. Health Risks and Musculoskeletal Problems of Elite Mobile Esports Players: a Cross-Sectional Descriptive Study. Sports Medicine-Open, 8(1), pp.1-9.

-          Please consider Gamers’ Thumb and Text Neck syndrome in the mapping of musculoskeletal problems.

-          Mobile e-athletes had significantly inferior spinal mobility and stability, as reference in:

Lam, W.K., Chen, B., Liu, R.T., Cheung, J.C.W. and Wong, D.W.C., 2022. Spine Posture, Mobility, and Stability of Top Mobile Esports Athletes: A Case Series. Biology, 11(5), p.737.

-          For VR headset, please consider cybersickness

-          For mental health, e-athletes and traditional athletes also faced the problem of stress because of competition. There was also abundant of resource on that.

-          The mental paragraph section is missing; especially psychosocial factors are important aspects. There were review papers on psychosocial factors on esports and some already cited by the authors. Please consider produce a data synthesis on this topic.

Singh, P., Sharma, M.K. and Arya, S., 2022. Esports and traditional sports players: An exploration of psychosocial profile. https://doi.org/10.21203/rs.3.rs-1907986/v1

 The authors may also consider discuss the similarities between esports and physical sports in some place of the paper. For example, they have stress before competition. They have problems about short career life. Both of them have “physical fitness training” on top of their “esports/sports training”.

Author Response

Please find attached our responses to your valuable comments

Reviewer 2 Report

Dear authors,

I have read the manuscript with interest. E-sports are a new topic for me, and I believe your article gave good insight in the physical demands and possible consequences of being a professional e-athletes. 

I have some (minor) comments which I hope might further improve your manuscript.

First the title: I believe your title: ‘E-sports: new athletes for old injuries’ does not cover the whole content of the manuscript. You do not only give insight in what injuries can occur among these ‘new’ athletes, but also provide information about what should be or can be done to protect them. In your conclusion you mention that e-athletes should be evaluated through a complete pre-participation evaluation and work on a correct training plan. I suggest that you try to incorporate this in your title, because this is your (most) important message to your readers, right? And if so, please re-read your manuscript and when apprpriate, be more explicite about what is necessary to protect the health of e-athletes.

Line 29, there is an extra space after ‘, and a..

Line 60/61: I prefer the aim that is included in the abstract. This aim is more extensive.

Table 1: check the outline of the table please.

Line 322/323: please check this sentence

Author Response

(The authors gave the same response as above.)
